

# Prevalence of bone loss surrounding dental implants as detected in cone beam computed tomography: a cross-sectional study

Fahda N. Algahtani[1], Mamata Hebbal[2], Moath M. Alqarni[3], Rahaf Alaamer[4], Anwar Alqahtani[4], Rahaf A. Almohareb[5], Reem Barakat[6] and Manal M. Abdlhafeez[7,8]

[1] Department of Clinical Dental Sciences, College of Dentistry, Princess Nourah bint Abdulrahman University, Riyadh, Saudi Arabia
[2] Department of Preventive Dental Sciences, College of Dentistry, Princess Nourah Bint Abdulrahman University, Riyadh, Saudi Arabia
[3] Dental Department, King Saud Medical City, Riyadh, Saudi Arabia
[4] College of Dentistry, Princess Nourah Bint Abdulrahman University, Riyadh, Saudi Arabia
[5] Department of Clinical Dental Sciences, College of Dentistry, Princess Nourah Bint Abdulrahman University, Riyadh, Saudi Arabia
[6] Dental Clinics Department, King Abdullah bin Abdulaziz University Hospital, Princess Nourah Bint Abdulrahman University, Riyadh, Saudi Arabia
[7] Department of Conservative Dental Sciences, College of Dentistry, Qassim University, Qassim, Saudi Arabia
[8] Department of Endodontics, Faculty of Dentistry, October University for Modern Sciences and Arts, 6th of October City, Egypt

Corresponding author
Rahaf A. Almohareb,
raalmohareb@pnu.edu.sa

## ABSTRACT

**Objectives.** The objective of this study was to assess the prevalence of crestal, and apical bone loss (CBL & ABL) associated with dental implants in CBCT scans. The second objective was to assess the radiographic stage of implant disease and the visible predisposing factors.

**Materials and Methods.** The CBCT scans that were taken from January 2015 to January 2022 in King Saud Medical City were screened to examine the marginal and periapical condition of dental implants. Information related to demographic variables, stage of bone loss, and radiographically evident predisposing factors were collected. The results were analyzed using descriptive statistics, chi-square test, and logistic regression analysis.

**Results.** In total, 772 implant scans were analyzed. The prevalence of crestal bone loss and apical bone loss around the implants were 6.9% and 0.4% respectively. The amount of bone loss was moderate in 52.8% of cases of CBL and 100% mild in cases of ABL. The risk factors for CBL were patient age ($p < 0.001$), implant location ($p < 0.001$), bone loss in proximal teeth ($p < 0.001$), and adjacent edentulous sites ($p < 0.001$). The risk factors for ABL were adjacent periapical infection ($p < 0.001$) and endodontic therapy ($p = 0.024$).

**Conclusion.** The prevalence of CBL and ABL was low. The CBCT can be used as a diagnostic tool for studying the prevalence of bone loss associated with peri-implant disease and relevant risk factors. The implantation of CBCT to evaluate the success and the prognosis of dental implants or the treatment of peri-implant diseases can be further considered in future research.

## INTRODUCTION

Peri-implantitis (PI) is an inflammatory process occurring in the tissues surrounding dental implants that results in progressive bone loss (*Carranza et al., 2018*). As classification of PI as a disease is controversial, its precise prevalence cannot be calculated (*Dreyer et al., 2018*). According to a recent systematic review, the prevalence of PI diversified from 0.4% in 3 years to 43.9% in 5 years (*Dreyer et al., 2018*). The etiology of PI is ill-defined (*Alani, Kelleher & Bishop, 2014*), though most sources consider PI a multifactorial disease, with causes including pathogenic bacteria/biofilms, exogenous irritants, and iatrogenic factors (*Sarmiento, Norton & Fiorellini, 2016*).

Periapical peri-implantitis (PPI) represents a recent manifestation of PI (*Alani, Kelleher & Bishop, 2014*). It was first described in a case report published in 1992, which discussed implants associated with a radiolucency exhibited only in the periapical regions in the absence of other pathological features, such as probing depths or marginal bone loss (*McAllister, Masters & Meffert, 1992*). PPI is frequently associated with such symptoms as pain, swelling, sinus tracts, and tenderness (*Quirynen et al., 2005*). However, limited data exist regarding the prevalence of PPI (*Alani, Kelleher & Bishop, 2014*). A retrospective study on PPI reported a low prevalence of 1.6% to 2.7% (*Quirynen et al., 2005*). To date, no published consensus concerning the exact etiology of PPI exists (*Di Murro et al., 2021*). However, several factors are thought to be related, such as the presence of endodontic infection in neighboring teeth, overheating of the bone during implant placement, residual pathology after tooth extraction, proximity of implants to infected maxillary sinuses, and compromising medical conditions, such as uncontrolled diabetes (*Alani, Kelleher & Bishop, 2014*; *Sarmast et al., 2016*; *Di Murro et al., 2021*).

Radiographically visible crestal bone loss is rarely evident in the absence of inflammatory signs and symptoms (*Berglundh et al., 2018*; *Caton et al., 2018*). Crestal bone loss (CBL) thus represents a key indicator of PI (*Caton et al., 2018*). Additionally, development of apical bone loss (ABL) constitutes a radiographic sign of PPI (*Shah et al., 2016*; *Caton et al., 2018*). However, periapical bone loss can also develop from overheating of the bone during surgical drilling for implants or from surgical drilling that is inappropriately deep relative to implant length (*McAllister, Masters & Meffert, 1992*; *Quirynen et al., 2005*). There is a shortage of studies assessing dental implant-associated crestal and apical bone loss via cone-beam computed tomography (CBCT) scans.

CBCT can provide information about the location of implants within alveolar bone, as well as their relationship to adjacent anatomical structures. Moreover, CBCT can provide information regarding bone defects that are not visible in periapical radiographs (*Song et al., 2021*). CBCT was found to be superior to conventional radiography in diagnosing bony defects of PI (*Song et al., 2021*). Since the bone loss associated with dental implants and some possible risk factors that could be only visible in CBCT were rarely studied using

CBCT imaging. The current cross-sectional study aimed to assess the prevalence of crestal and apical bone loss associated with dental implants using CBCT scans. This study further aimed to evaluate the radiographic stages of implant disease and assess radiographically observable predisposing factors.

## MATERIAL AND METHODS

### Study sample

This study was submitted for ethical approval to the institutional review board of Princess Nourah Bint Abdulrahman University (PNU) (21-0499) and King Saud Medical City (KSMC) (H1RI-04-Sep22-04). The study was conducted in King Saud Medical City in Riyadh, Saudi Arabia. King Saud Medical City is the main public medical complex of the Ministry of Health (MOH) that serves Riyadh and adjacent small cities. The authors determined the sample size following a former study concerning the prevalence of PI (*Aljasser et al., 2021*). The following formula was used to calculate sample size:

$$n = (z)^2 \, p \, (1-p) / d^{2*}$$
$$n = (1.96)^{2*}23.76*74.26/3*3$$
$$n = 753 \text{ CBCT scans}$$

where n is the sample size, z is the level of confidence based on the standard normal distribution (z for 95% = 1.96), p is the prevalence based on the reference study ($p = 23.76$), and d is the allowed margin of error (d = 3%).

All patients treated in KSMC were asked to sign general treatment consent forms that permit the release of health information for educational or research purposes while concealing their personal identities. All CBCT scans taken in the King Saud Medical City radiology department between January 2015 and January 2022 were screened to examine the marginal and periapical conditions of dental implants. These scans were taken for the diagnosis and management of oral and dental health conditions. A qualified technician used a *CS 9300* 3D digital imaging system (Carestream, Rochester, NY, USA) to capture the radiological images in accordance with the manufacturer's recommendations. The CBCT scans fields of view (FOV) ranged from $8 \times 8$ cm to $10 \times 10$ cm were included in the study. Tube voltage was 90 KV, tube current was 5 mA, exposure time ranged from 12–20 s, and resolution ranged from 0.18–0.3 mm. The purpose of these scans was unrelated to the purpose of the current study. The inclusion criteria were CBCT scans that showed dental implants in patients who were 18 years of age and older. In the presence of more than one scan per patient, the most recent scan was included in the study. The exclusion criteria were low-quality CBCT scans (such as those that had artifacts).

### Radiographic examination

The total number of patients who underwent CBCT scans during this period (January 2015 to January 2022) was determined using the dental hospital's electronic records system. Images were accessed and assessed using the hospital's Carestream Dental 3D Imaging Software (Rochester, NY, USA).

Demographic data included patients' gender and age at the time of the scan. CBCT scans were viewed by one examiner (a periodontist) in the same setting in which they were

recorded. To assess intra-observer reliability, the intraclass correlation coefficient (ICC) was calculated according to two continuous variables—length of the dental implant located above the alveolar bone crest (1 mm) and distance from the adjacent tooth (1 mm). An ICC of 0.90 was considered acceptable for this study. The ICC was calculated for 30 samples and was found to be 96% for the length of the dental implant located above the alveolar bone crest and 99% for the distance of the implant from the adjacent tooth.

The examiner had one research assistant who collected the data presented in Table 1 using REDCap software (Nashville, TN, USA). The research assistant recorded the data for all implants present in the dental arch. Crestal bone loss (CBL) was diagnosed when 3 mm of the dental implant threads were located above the alveolar bone crest (*Fransson et al., 2005*; *Fransson, Wennström & Berglundh, 2008*). Meanwhile, periapical bone loss (PBL) was identified when there was localized apical radiolucency surrounding the implant apex that was distinct from marginal bone loss (*Shah et al., 2016*).

The stage of CBL was determined based on *Froum & Rosen (2012)*'s classification. Measurements began at the implant shoulder and continued until the lesion confined apically (Figs. 1 and 2). The derived values were then calculated as percentages using the following formula: *(distance from implant shoulder to apical extent of CBL)/total implant length) × 100.*

The CBL loss was then categorized as follows (Fig. 1):

1. Stage I (early): bone loss < 25% of implant length;

2. Stage II (moderate): bone loss of 25–50% of implant length;

3. Stage III (advanced): bone loss > 50% of implant length.

Furthermore, CBL was classified according to the shape of the bone defect into crater-like bone defects, infra-bony defects, and dehiscence (Fig. 1) (*Song et al., 2021*).

Classification of ABL was performed according to *Shah et al. (2016)* Measurements of bone loss, in millimeters, began at the implant apex and progressed coronally (Figs. 1, 2). These values were then converted to percentages using the following formula: *(distance of bone loss from implant apex to coronal extent of PBL/total implant length) ×100.*

Based on the resulting percentages, ABL was classified into three groups (Fig. 1):

1. Class I indicates mild lesions. In these cases, radiographic bone loss accounts for less than 25% of the implant length;

2. Class II indicates moderate lesions. In these cases, radiographic bone loss accounts for 25–50% of the implant length;

3. Class III indicates advanced radiographic bone loss that accounts for more than 50% of the implant length.

## Data analysis

The data were collected using REDCap and an Excel sheet of the collected data was generated. Data analysis was performed using SPSS (IBM SPSS Statistics for Windows, Version 21.0. Armonk, NY, USA). Descriptive statistics were calculated in terms of frequency and percentages for categorical data. Chi-square was used to examine the relationship between CBL/ABL and patients' demographics and radiographically visible

**Table 1** The data collection instrument.

1–**Gender:** Male-Female

2–**Patient age at the time of scan (The patient age according to DOB minus time of CBCT scan)**

3–**Implant Location:** Maxillary anterior region –maxillary posterior region- mandibular anterior region- mandibular posterior region

4–**Condition of the adjacent tooth: (based on worst condition) (multiple select allowed)**

Normal –missing–endodontic therapy (previously accessed or treated) –periapical lesion –periodontal disease

5–**Type of prosthesis:** Single crown –multiple units –no prosthesis

6–**Implant angulation/ alignment:** well centered –proximally inclined (mesial or distal) - axial inclination (buccal or lingual) -Proximal and axial inclination.

7–**Implant position/center within the bone:** too buccal (less than 1.5 buccal bone)–too lingual (less than 1.5 lingual bone) –well centered - not surrounded by bone.

8–**Distance of the implant to the adjacent tooth: (based on the closest tooth)** Less than (2 mm) –(2 mm) –more than 2 mm –not applicable

9–**Length of the dental implant located above the alveolar crest (in mm):** (comment field)

10–**Peri-implant bone health condition:** Healthy –crestal bone loss–apical bone loss

11–**If the implant had crestal bone loss, select the stage of crestal bone loss:**

Early (bone loss less than 25% of the implant length.)

Moderate (bone loss 25% to 50% of the implant length)

Advanced (bone loss of more than 50% of the implant length)

12–**Select the shape of the crestal bone defect:**

Dehiscence
Infra-bony defect
Crater like defect

13–**Locally visible predisposing factors for crestal bone loss:**

The implant is positioned too buccally or too lingually.

The implant is placed in proximity of the adjacent teeth.

Implant malalignment

Radiographically visible cement remnant

Radiographically visible calculus

No radiographically apparent reason.

Other. (Comment field)

14–**If the implant had periapical bone loss, select the stage of periapical bone loss:**

**Class I mild** (bone loss <25% of implant length from implant apex).

**Class II moderate** ((bone loss 25–50% of implant length from implant apex),

**Class III advanced** ((bone loss 25–50% of implant length from implant apex),

15–**Locally visible predisposing factors for periapical bone loss:**

The implant apex approached the apex of an adjacent tooth.

The implant apex became infected by adjacent periapical pathology on the adjacent tooth/implant.

The implant apex was placed angulated (lingually/labially) outside the bone envelope.

Implant apex approached a proximal anatomical region (specify in the comment field (maxillary sinus –nasal cavity –nerve))

No radiographically apparent reason.

Other. (Comment field)

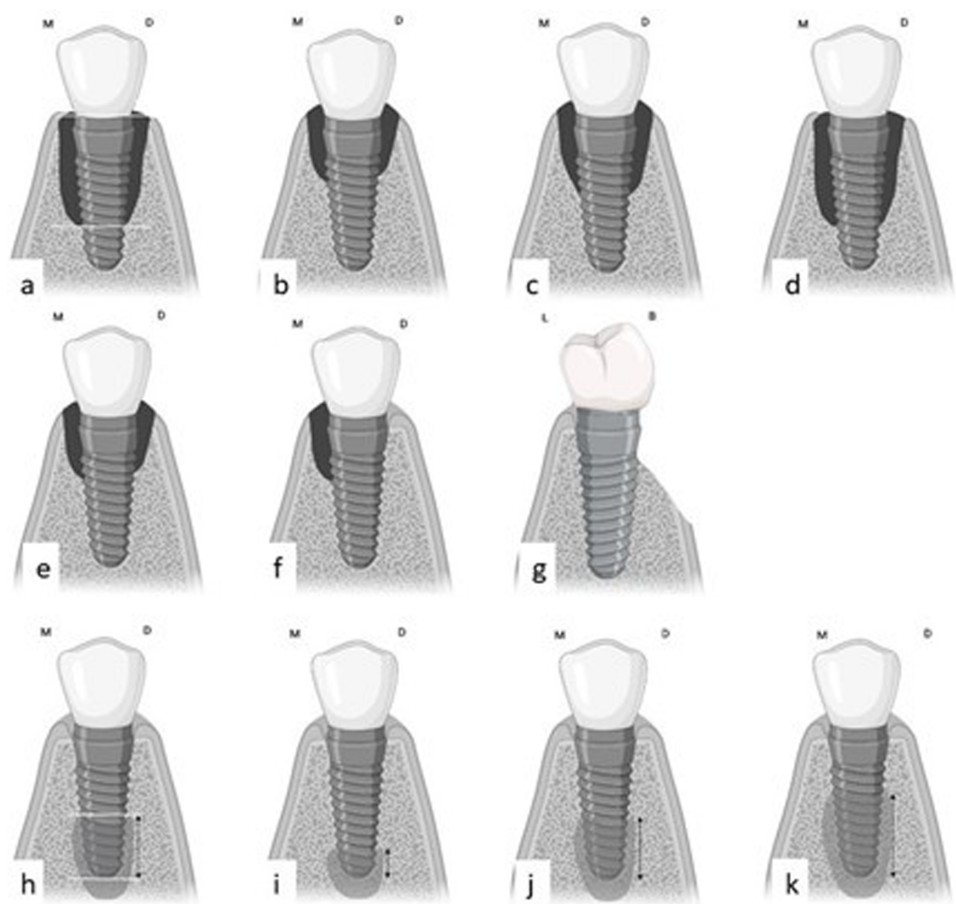

**Figure 1** **The method used to measure and classify CBL and ABL.** (A) The method used for measuring CBL starts from the distance from the implant shoulder to the most apical confines of the lesion. (B, C, D) The stage of CBL loss started with: (B) Early-stage I lesion (% of bone loss is <25%); (C) Moderate-stage II lesion (% of bone loss is 25%–50%); (D) Advanced-stage III lesion (% of bone loss is >50%). (E, F, G) Shapes of peri-implant bone defects: (E) crater-like defect; (F) Infra-bony defect; (G) Dehiscence. (H) The method used for measuring the ABL starts from the implant apex till the most coronal confines of the lesion; (I, J, K) The classes of ABL: (I) Class I lesion (% of bone loss is <25%); (J) Class II lesion (% of bone loss is 25%–50%); (K) Class III lesion (% of bone loss is >50%). Created with Biorender.com.

local predisposing factors. The confidence interval was 95% (CI) and the significant level was set at $p < 0.05$.

## RESULTS

The study included 772 implants. Table 2 shows the distribution of study samples according to the presence and absence of CBL and ABL in relation to variables like gender, age, region of implant placement, condition of the adjacent tooth, and implant status. There were 462 female and 310 male patient records. The mean age of the sampled patients was 48.87 years. A total of ninety-seven implants (12.6%) were present in the anterior maxillary region, 291 implants (37.7%) were present in the posterior maxillary region, eighty-one implants

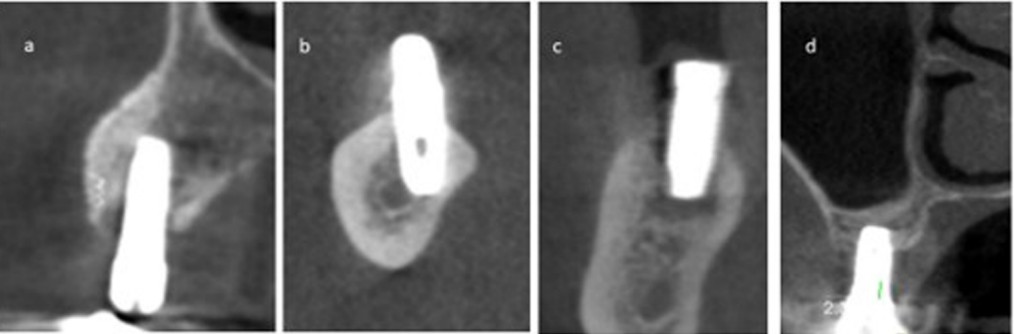

**Figure 2  The coronal view of a few CBCT scans that show the extent of CBL.** (A) Early-stage I CBL; (B & C) Moderate-stage II CBL; (D) Advanced-stage III CBL.

(10.5%) in the mandibular anterior region, and the other 303 implants (39.2%) were in the mandibular posterior region. A total of 53 (6.9%) and 3 (0.4%) implants have CBL and ABL, respectively.

Single prostheses were present in 98 implants (12.7%), multiple unit prostheses were present in 103 implants (13.3%), and the rest of the implants (74%) had no prostheses. Regarding implant angulation and position within the bone, 519 implants (67.2%) were centrally angulated within the bone, 84 implants (10.9%) were proximally inclined, 142 implants (18.4%) were axially inclined, and 27 implants (3.5%) had both proximal and axial inclination. The majority of implants (93.1%) were well-centered within the bone, while the remainder were either too buccal (4.1%) or too lingual (2.7%).

The bivariate analysis was used to study significant relationships. There was no statistically significant difference in the CBL to gender ($p = 0.618$). There was a statistically significant difference in CBL according to age, region of implant placement, and periodontal condition of the adjacent tooth ($p < 0.001$). There was a significant association between CBL and implants placed among the older age group individuals (16.2% and 13.3% in the categories 60–69 and 70+ years, respectively). Moreover, implants in the mandibular anterior region (24.7%), and with an adjacent tooth having periodontal disease (50%) or a missing adjacent tooth (11.1%) had a significant association with CBL. Types of implant prosthesis, implant angulation and position within the bone did not reveal any statistical significance to the presence of CBL, whereas the distance of the implant to the adjacent tooth showed a statistically significant relationship ($p < 0.001$). The presence of periapical lesion in the adjacent tooth ($P < 0.001$) and endodontic therapy ($P = 0.024$) were significantly associated with ABL.

In terms of disease staging, the majority of implants with CBL (52.8%) were graded as stage II, indicating moderate bone loss (Fig. 2). The type of bone defect was crater-like bone defects in the majority of cases (41 implants, 77.4%). Followed by a dehiscence-shaped bony defect (11 implants, 20.8%) while infra-bony defects (1, 1.9%) affected the least number of cases. All the implants with ABL (100%) were scored as class I, indicating mild

**Table 2  Distribution of study samples according to CBL/ABL, demographics, and implant-related factors.** Bivariate analysis (chi-square test) was used to study associations and possible risk factors for CBL and ABL.

| | | Crestal bone loss (CBL) | | | Apical bone loss (ABL) | | |
|---|---|---|---|---|---|---|---|
| | | Healthy No (%) | CBL No (%) | p value | Healthy No (%) | ABL No (%) | p value |
| | Total (772) | 719 (93.1) | 53 (6.9) | | 769 (99.6) | 3 (0.4) | |
| **Gender** | Male | 287 (92.6) | 23 (7.4) | 0.618 | 309 (99.7) | 1 (0.3) | 0.809 |
| | Female | 432 (93.5) | 30 (6.5) | | 460 (99.6) | 2 (0.4) | |
| | 20–29 | 55 (98.2) | 1 (1.8) | | 56 (100) | 0 (0) | |
| | 30–39 | 153 (95) | 8 (5) | | 161 (100) | 0 (0) | |
| **Age** | 40–49 | 146 (96.1) | 6 (3.9) | <0.001 | 151 (99.3) | 1 (0.7) | 0.701 |
| | 50–59 | 228 (95) | 12 (5) | | 238 (99.2) | 2 (0.8) | |
| | 60–69 | 124 (83.8) | 24 (16.2) | | 148 (100) | 0 (0) | |
| | >70 | 13 (86.7) | 2 (13.3) | | 15 (100) | 0 (0) | |
| | Maxillary Anterior | 95 (97.9) | 2 (2.1) | | 96 (99) | 1 (1) | |
| **Arches and region** | Maxillary Posterior | 287 (98.6) | 4 (1.4) | <0.001 | 290 (99.7) | 1 (0.3) | 0.707 |
| | Mandibular Anterior | 61 (75.3) | 20 (24.7) | | 81 (100) | 0 (0) | |
| | Mandibular Posterior | 276 (91.1) | 27 (8.9) | | 302 (99.7) | 1 (0.3) | |
| **Condition of adjacent tooth** | Tooth present | 367 (97.6) | 9 (2.4) | <0.001 | 374 (99.5) | 2 (0.5) | 0.533 |
| | Tooth missing | 352 (88.9) | 44 (11.1) | | 395 (99.7) | 1 (0.3) | |
| **Condition of adjacent tooth (Periodontal disease)** | Absent (Missing teeth included n-396) | 717 (93.4) | 51 (6.6) | <0.001 | 765 (99.6) | 3 (0.4) | 0.900 |
| | Present | 2 (50) | 2 (50) | | 4 (100) | 0 (0) | |
| **Condition of adjacent tooth (Periapical lesion)** | Absent (Missing teeth included n-396) | 702 (93.2) | 51 (6.8) | 0.523 | 751 (99.7) | 2 (0.3) | <0.001 |
| | Present | 17 (89.5) | 2 (10.5) | | 18 (94.7) | 1 (5.3) | |
| **Type of implant prosthesis** | No prosthesis/ Single crown | 135 (96.4) | 5 (3.6) | 0.088 | – | – | - |
| | Multiple units | 584 (92.4) | 48 (7.6) | | – | – | |
| | Centered | 486 (93.6) | 33 (6.4) | | 486 (93.6) | 33 (6.4) | |
| **Implant angulation** | Proximal inclination | 80 (95.2) | 4 (4.8) | 0.082 | 80 (95.2) | 4 (4.8) | 0.082 |
| | Axial inclination | 131 (92.3) | 11 (7.7) | | 131 (92.3) | 11 (7.7) | |
| | Proximal & Axial inclination | 22 (81.5) | 5 (18.5) | | 22 (81.5) | 5 (18.5) | |
| | Well centered | 668 (92.9) | 51 (7.1) | | – | – | – |
| **Implant position within bone** | Too buccal | 32 (100) | 0 (0) | 0.266 | – | – | |
| | Too lingual | 19 (90.5) | 2 (9.5) | | – | – | |
| | <2 mm | 59 (98.3) | 1 (1.7) | | – | – | – |
| **Distance of implant to adjacent tooth** | 2 mm | 28 (93.3) | 2 (6.7) | 0.334 | – | – | |
| | >2 mm | 284 (97.6) | 7 (2.4) | | – | – | |

bone loss (Fig. 3). A few CBCT scans that show the extent of bone loss in cases of CBL and ABL are shown in Figs. 2 and 3.

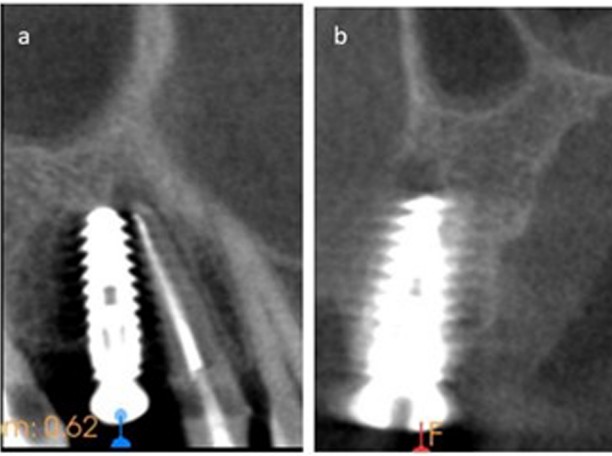

**Figure 3** **The CBCT scan shows one case of ABL in two different planes.** (A) The sagittal plane shows the apex of the previously treated first premolar that has periapical radiolucency approaching the apex of the implant in the adjacent site. (B) The coronal view shows mild bone loss that is designated for Class I ABL.

## DISCUSSION

The 772 dental implants that were individually evaluated in CBCT scans revealed a prevalence of 6.9% CBL and 0.4% APL. The factors that were significantly associated with CBL were patient age, implant location, evidence of bone loss or periodontitis in the adjacent teeth, and edentulous sites proximal to the dental implant. The shapes of most of these crestal defects were crater-like and dehiscence, and they were least likely to be infra-bony. However, periapical lesions and endodontic therapy near the dental implant were linked to ABL.

A dental radiograph is an essential tool for the diagnosis of PI and PPI (*Berglundh et al., 2018*). In particular, CBCT was described as the most sensitive tool for CBL surrounding dental implants (*Song et al., 2021*). The CBCT allows visualization of the amount of bone loss and the shape of the bony defect (*Song et al., 2021*). Bone loss surrounding the dental implant is an indicative sign of PI, which rarely happens in the absence of soft tissue inflammation (*Schwarz et al., 2018*). PI was found to be prevalent in 10% of the implant cases and 20% of the implant patients (*Mombelli, Müller & Cionca, 2012*). On the contrary, PPI is diagnosed as radiolucency surrounding the apex of dental implants (*Schwarz et al., 2018*). The prevalence of PPI was described as rare and was found to be in 0.34% of dental implants (*Di Murro et al., 2021*). Clinical diagnosis could have supported the diagnosis of PI or PPI in this study, especially since PI usually develops five to ten years after implant placement (*Mombelli, Müller & Cionca, 2012*). However, the study design was a cross-sectional design of CBCT radiographs. Therefore, the prevalence of CBL and ABL was studied rather than the clinical finding of PI. The advantage of this unique CBCT-based cross-sectional study was its simplicity in comparison to case-based and cohort studies (*Levin, 2006*). Moreover, the findings of this study can be used as a

foundation for future cohort studies (*Levin, 2006*). The use of CBCT to study bony defects associated with dental implants offered a great opportunity to examine their prevalence and radiographic classification. Moreover, it provided information regarding the effect of locally predisposing factors such as implant position and alignment on peri-implant bone health.

CBL associated with PI usually develops after five to ten years of implant placement (*Mombelli, Müller & Cionca, 2012*). This could partially explain the prevalence of CBL in the older age group. The same findings were reported in a retrospective study; however, a systemic analysis concluded that age was not a risk factor for developing PI (*Renvert et al., 2014*; *Dreyer et al., 2018*). The same review found that PI developed as the function time exceeded five to ten years (*Dreyer et al., 2018*). Periodontitis was another risk factor for PI, which could further confirm the findings of this study (*Dreyer et al., 2018*). Moreover, a systemic review and meta-analysis found that implant locations were a significant factor in developing PI (*Song et al., 2020*). The most common sites were the mandibular anterior regions, followed by the maxillary anterior and mandibular posterior regions, which had approximate risk ratios (*Song et al., 2020*). Similarly, the findings of this study indicated that the most common site for CBL was the mandibular anterior region, followed by the mandibular posterior region and the maxillary anterior region. In both studies, the maxillary posterior region was the least common site for CBL (*Song et al., 2020*). The findings of this study suggested that CBL is more evident in implant sites adjacent to edentulous areas compared to natural teeth. An explanation for this incident is that edentulous sites might represent areas where teeth were lost due to periodontal disease. Since periodontitis is a common risk factor for CBL, however, this assumption was not investigated and could not be confirmed in this study (*Dreyer et al., 2018*). The shape of the CBL bony defect could have a potential impact on the success of peri-implant constructive surgery (*Tomasi et al., 2019*). For example, the presence of four wall defects was among the most predictable outcomes for reconstructive surgeries (*Aghazadeh, Persson & Renvert, 2020*).

The study found that ABL was associated with the known risk factors for periapical peri-implantitis (*Alani, Kelleher & Bishop, 2014*; *Sarmast et al., 2017*; *Di Murro et al., 2021*). The presence of adjacent periapical infection, particularly in root canal-treated teeth, was one of these risk factors (*Alani, Kelleher & Bishop, 2014*; *Sarmast et al., 2017*; *Di Murro et al., 2021*). Possible alternative causes are bone overheating and drilling, which were not investigated in this study (*Alani, Kelleher & Bishop, 2014*; *Di Murro et al., 2021*). However, PPI's histological and microbiological findings suggested residual endodontic or periodontic infection from adjacent teeth or extraction sites (*Marshall et al., 2019*). Fortunately, the literature suggested that the healing of PPI was evident after the endodontic intervention followed by surgical debridement and grafting of the infected implant (*Sarmast et al., 2017*).

The limitations of the study are the same as those of any cross-sectional design study. Unlike longitudinal studies, a cross-sectional study design is created to study both the outcome and risk factors at a single point in time (*Levin, 2006*). Therefore, it does not provide an opportunity to study casual relationships (*Levin, 2006*). However, it provides

information on risk factors associated with a particular disease and serves as a foundation for the synthesis of the hypothesis (*Levin, 2006*). The CBCT in this study was a helpful tool to study the prevalence of bone loss surrounding dental implants and possible risk factors for CBL such as implant position within the arch. The CBCT also provided information regarding the type of bony defect which could be helpful for categorizing peri-implant disease and their response for possible intervention. However, the assessment of bone quality in terms of bone density can be limited in CBCT imaging due to various factors such as scattered radiation and current techniques in the reconstruction of imaging algorithms (*Pauwels et al., 2015*). To overcome this limitation, current research and clinical applications have shifted from the assessment of bone density using the Hounsfield index to the assessment of bone quality using structural analysis (*Pauwels et al., 2015*). It's difficult to standardize Hounsfield analysis over a large number of scans. In this study, attempts were made to assess bone quality in CBCT scans derived from singular CBCT machines and in selected FOVs. Another limitation of CBCT imaging is the beam hardening phenomenon that compromises dense bone evaluations, especially in the presence of dental implants (*Nagarajappa, Dwivedi & Tiwari, 2015*). Multidetector computed tomography (MDCT) could overcome this limitation and improve the accuracy of the evaluation of metallic dental implants (*Draenert et al., 2007*). CBCT scans with artifacts were excluded from the study because they provided limited valuable information. Nevertheless, CBCT imaging provides valuable information since it allows for the three-dimensional evaluation of bony structures. CBCT imaging is now a valuable tool, and it was incorporated in treatment planning, management, and evaluation of treatment outcomes in many fields of dentistry (*Alamri et al., 2012*). Therefore, future clinical cohort studies that use CBCT to evaluate the health of the peri-implant are recommended. Moreover, a comparison between CBCT and MDCT in the evaluation of bone quality, especially in the presence of extensive metallic restoration and dental implant can be further examined in future research.

## CONCLUSIONS

Within the limitations of this study, the prevalence of CBL and ABL was 6.9% and 0.4%, respectively. Most of the bony defects associated with dental implants were crater-like defects, followed by dehiscence and infra-bony defects. The stage of detected bone loss was early to moderate in cases of CBL and mild in cases of ABL. Patient age, implant location, evidence of bone loss in adjacent teeth, and presence of an adjacent edentulous area were the most significant factors associated with CBL, while patient gender, type of prosthesis, implant angulation, and position within the arch had no significant relationship with the occurrence of CBL. The presence of periapical lesions or endodontic treatment adjacent to the dental implant was associated with the occurrence of ABL.

## Funding

This original research was funded by the Princess Nourah bint Abdulrahman University Researchers Supporting Project number (PNURSP2023R363), Princess Nourah bint Ab-dulrahman University, Riyadh, Saudi Arabia. The funders had no role in study design, data collection and analysis, decision to publish, or preparation of the manuscript.

## Grant Disclosures

The following grant information was disclosed by the authors:
Princess Nourah bint Abdulrahman University Researchers Supporting Project number: PNURSP2023R363.
Princess Nourah bint Ab-dulrahman University, Riyadh, Saudi Arabia.

## Competing Interests

The authors declare that there are no competing interests.

## Author Contributions

- Fahda N. Algahtani conceived and designed the experiments, authored or reviewed drafts of the article, and approved the final draft.
- Mamata Hebbal conceived and designed the experiments, analyzed the data, authored or reviewed drafts of the article, and approved the final draft.
- Moath M. Alqarni conceived and designed the experiments, prepared figures and/or tables, and approved the final draft.
- Rahaf Alaamer performed the experiments, authored or reviewed drafts of the article, and approved the final draft.
- Anwar Alqahtani performed the experiments, authored or reviewed drafts of the article, and approved the final draft.
- Rahaf A. Almohareb conceived and designed the experiments, prepared figures and/or tables, and approved the final draft.
- Reem Barakat conceived and designed the experiments, prepared figures and/or tables, and approved the final draft.
- Manal M. Abdlhafeez performed the experiments, prepared figures and/or tables, and approved the final draft.

## Data Deposition

The raw measurements are available in the Supplemental Files.

## Supplemental Information

Supplemental information for this article can be found online at http://dx.doi.org/10.7717/peerj.15770#supplemental-information.

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
