# Peer review of "Prevalence of bone loss surrounding dental implants as detected in cone beam computed tomography: a cross-sectional study"

_PeerJ, doi:10.7717/peerj.15770_

## Round 0.1 · original submission · Minor Revisions

Dear authors,

Kindly take into consideration all of the reviewer's suggestions and make the necessary changes accordingly.

Sincerely,

·

Basic reporting

Dear Authors,

I have read your study entitled "Prevalence of Bone Loss Surrounding Dental Implants as Detected in Cone Beam Computed Tomography: A Cross-Sectional Study." However, I have a few questions and suggestions that could improve the manuscript's clarity. Please consider the following points:

I suggest referencing the following statement: "Since the classification of IP as a disease is controversial, its precise prevalence cannot be calculated."
It would be beneficial to include a description of why the chosen method was used.
I recommend including the inclusion and exclusion criteria in the methodology section.
I suggest incorporating references to more recent articles in the discussion section.
It would be valuable to include the following limitation of the study: the difficulty of performing a standardized Hounsfield analysis when dealing with a large number of exams.
I suggest mentioning in the study that the analysis of bone loss near a dense region is limited due to the potential presence of the Beam hardening phenomenon.
I recommend adding more current references. Overall, your study provides valuable data.
Please take these suggestions into consideration to enhance the clarity and organization of your manuscript. Thank you for your contribution to the field.

Sincerely,

Experimental design

no comment

Validity of the findings

no comment

Additional comments

no comment

·

Basic reporting

The article is good, brings some interesting information, but needs to be clarified in just a few points highlighted in the PDF. References have a good quality, and data is well organized.

Experimental design

The article fits well in the scope of the journal.

Validity of the findings

Just a few points to be observed in the PDF.

---

## Round 0.2 · accepted · Accept

Dear authors, thank you for choosing PeerJ to publish your article.